# Contributions to a Discussion of *Spinosaurus aegyptiacus* as a Capable Swimmer and Deep-Water Predator

**DOI:** 10.3390/life11090889

**Published:** 2021-08-28

**Authors:** Jan Gimsa, Ulrike Gimsa

**Affiliations:** 1Department of Biophysics, University of Rostock, Gertruden Str. 11A, 18057 Rostock, Germany; 2Research Institute for Farm Animal Biology (FBN), Institute of Behavioural Physiology, Wilhelm-Stahl-Allee 2, 18196 Dummerstorf, Germany; gimsa@fbn-dummerstorf.de

**Keywords:** dorsal sail, swim tail, head crest, pivot feeding, hydrodynamics, allometry, Reynolds number, submerged hunting

## Abstract

The new findings on *Spinosaurus’* swim tail strongly suggest that *Spinosaurus* was a specialized deep-water predator. However, the tail must be seen in the context of the propelled body. The comparison of the flow characteristics of *Spinosaurus* with geometrically similar animals and their swimming abilities under water must take their Reynolds numbers into account and provide a common context for the properties of *Spinosaurus*’ tail and dorsal sail. Head shape adaptations such as the head crest reduced hydrodynamic disturbance and facilitated stealthy advance, especially when hunting without visual contact, when *Spinosaurus* could have used its rostral integumentary mechanoreceptors for prey detection. The muscular neck permitted ‘pivot’ feeding, where the prey’s escape abilities were overcome by rapid dorsoventral head movement, facilitated by crest-mediated lower friction.

## 1. Introduction

Its discoverer, the German paleontologist Ernst Stromer (1871–1952), was perplexed by the large predatory dinosaur *Spinosaurus aegyptiacus* [1]. The fossil remains that he excavated in the Sahara were in a puzzling disproportion to the known anatomies of large predatory dinosaurs. Stromer found many remains of aquatic and swamp animals. Additionally, other large predators such as crocodilians made the areas unsafe, despite the low number of major terrestrial herbivores [2,3]. So, what did a large predator like *Spinosaurus* feed on?

The discussion on *Spinosaurus*’ swimming abilities probably began in 1984 with Taquet [4] and has intensified since 2014 when Ibrahim et al. [5] suggested a semi-aquatic lifestyle, similar to Amiot et al. [6], who argued that oxygen isotopes suggest that other spinosaurids that are closely related to *Spinosaurus*, were probably aquatic. Ibrahim et al. [5] specified adaptations such as *Spinosaurus*’ dense bone structure, which is known from dugongs, desmostylia all the way to penguins. The reduced buoyancy together with a forward position of the center of mass would have enabled better swimming and diving. In favor of swimming is *Spinosaurus*’ short pelvic girdle and hindlimbs and low, flat-bottomed pedal unguals, which were coincident with digital lobes or webbing in shore birds. The small nostrils were in a posterior position to inhibit the intake of water. The elongated snout featured more than 100 rostral neurovascular foramina, through which probably integumentary mechanoreceptors were connected with the complexly branched trigeminal nerve. They may have formed a passive “sonar” structure, which is also known from pliosauroids and extant crocodilians and would have enabled *Spinosaurus* to sense prey movements in water. Despite these and other adaptations, Ibrahim et al. [5] did not go so far as to propose submerged hunting but that “the flexibility of the tail and the form of the neural spines” in *Spinosaurus* suggests swimming with lateral undulations of the tail with the dorsal sail functioning as a display structure remaining visible while swimming.

Even before the new work of Ibrahim et al. [7], several new findings have strengthened our hypothesis [8] that *Spinosaurus* was a capable swimmer with the dorsal sail serving hydrodynamic purposes during submerged swimming, a snout morphology adapted for detecting, biting and grabbing evasive prey in aquatic settings [9] and the convergent evolution of *Spinosaurus*’ jaws and teeth with those of the predatory pike conger eel [10]. Clearly, the earlier interpretations that *Spinosaurus*’ dorsal spines supported either a humpback storage that would also collect and store heat [11] or a non-submersible dorsal sail that was used to give-off heat [12,13] became obsolete.

The view that *Spinosaurus* was a capable swimmer, however, was mostly questioned until now. Hone and Holtz [14] rejected the hypothesis that *Spinosaurus* had been an able swimmer in deep water because it is based on a number of so far unverifiable assumptions about the anatomy and locomotion of these animals. Henderson [15] criticized the lack of more quantitative studies in the form of hydrodynamical and biomechanical analyses. His own numerical buoyancy analysis concluded that *Spinosaurus* was unable to descend and would have been unstable when floating. However, this was published about two years before the swimming tail became known.

Following the lucky excavation of a large part of the fossil remains of the swim tail, the chances for a discussion of how *Spinosaurus* hunted under water are much improved. Clearly, the anatomy of the ‘new’ flexible tail makes its use as a hunting weapon similar to thresher sharks’ tails as proposed by us very unlikely. Additionally, the concept of crocodile-like horny scales, improving the efficiency of the undulating propulsion [8] became obsolete for *Spinosaurus* but may have been realized in closely related piscivorous species, such as *Ichthyovenator* [16], *Irritator* [17,18] or *Baryonyx* [19,20,21]. In our view, the strong similarity of *Spinosaurus*’ sail shape with the dorsal fin of sailfish suggests that of all spinosaurids, *Spinosaurus* had the best-adapted sail shape for an aquatic lifestyle. One can compare the dorsal sail to the centerboard of a sailboat. The submerged sail improved maneuverability and provided the hydrodynamic fulcrum for powerful tail movements during acceleration phases and rapid neck movements when grabbing prey. In the ensuing fight with larger prey, the sail would have provided hydrodynamic stability. However, this required a stable sail, able to absorb shocks and lateral bending forces, which were strongest at the base of the sail when counteracting rolling around the craniocaudal axis. We assume that corresponding adaptations are visible in the anatomy of the dorsal spines of the sail, such as the disc-shaped proximal flanges already described by Stromer [1], which could provide elasticity during lateral deflections of the sail.

Up to now, the discussion of the swimming abilities fell short of a common context for the properties of the ‘new’ tail and dorsal sail. In our view, conclusions on the hydrodynamic properties of *Spinosaurus* from crested newts [7] can be drawn only based on allometric principles, taking the animals’ sizes and velocities into account. Considerations on the hunting behavior must combine the swimming abilities with the head and snout adaptations and the ability to perform swift head movements, which is known to be a key factor for aquatic predators [22]. They must also explain the benefit of the assumed powerful forelimbs with the large thumb claws that seem ideal for hooking and slicing slippery prey caught with the snout, rather than catching prey or even digging [14].

Here, we propose that the long, flexible neck [23] and the specialized head and snout morphology with the ridged longitudinal fluted crest were all part of the adaptations to ‘pivot’ feeding, where rapid head movement overcomes the prey’s escape abilities as observed in seahorses [24]. Pivot feeding required stealthy advance to bring the prey into *Spinosaurus*’ reach without triggering an escape. Head shape adaptation, minimizing hydrodynamic disturbance and reducing friction, would have improved hunting success, especially in turbid and low-light aquatic environments, when *Spinosaurus* could have used rostral integumentary mechanoreceptors for prey detection like pike conger eels and crocodilians [10,25,26,27].

## 2. Methods

Here, we further develop our ideas on the submerged swimming and hunting abilities of *Spinosaurus*, first published in [8]. Based on our hypothesis that *Spinosaurus* was a capable swimmer, we predicted the largely correct silhouette of the tail, albeit assuming horny scales, before the fossil remains of the swimming tail were found [7]. In the present analysis, we mainly comment on newly published data and discussions about *Spinosaurus* in the literature, and by applying and reconciling knowledge from other scientific fields such as biophysics, biomechanics of fish swimming, hydrodynamics, etc., to draw a concise picture of *Spinosaurus* ethology. However, our new hypotheses will not be easy to prove. It should be checked whether they are compatible with new fossil evidence and hydrodynamic studies of extinct and extant creatures.

In our view, the current discussion of *Spinosaurus* lacks a common context for the characteristics of the tail and dorsal sail, as well as for the size of the animal. This is a critical point in the interpretation of the data published by [7], which bypassed the discussion of the role of the dorsal sail. Our ideas may help shed light on the hydrodynamic and ethological aspects of *Spinosaurus*. We believe that coherent conclusions about (semi-) aquatic animals can only be drawn based on allometric principles considering their Reynolds numbers.

## 3. Discussion

### 3.1. Buoyancy and Balance Aspects

Ibrahim et al. [7] calculated a volume of 3.864 m^3^ for their 10-m model of a subadult *Spinosaurus*, obtaining a mass range of 3219–4173 kg for a variety of reasonable assumptions about tissue densities and the volume of air-filled spaces. The resulting wide range of 0.833 to 1.070 kg/L indicates a great uncertainty in calculating the overall body densities in common floatation models. For a semiaquatic life style, an optimal density would probably be slightly below that of fresh-water of 1 kg/L when floating on the surface. When diving, negative buoyancy would be assured by compression of the air-filled compartments. In the numerical models of Henderson [15] and Ibrahim et al. [7]**,** this condition can be fulfilled by a number of reasonable assumptions, such as low air-sac volumes and a skeleton with a higher bone density. To reduce buoyancy by decreasing oxygen storage in the lungs for diving excursions, oxygen can be stored in body fluids and tissues, as in diving birds such as penguins [28,29] or in seals and whales, which have high amounts of the dense oxygen-storing protein myoglobin in their muscle tissues [30,31]. A straight-forward way of reducing buoyancy would be to assume the presence of gastroliths, which are found in many extant semiaquatic animals, even axolotls, and have been common in plesiosaurs and sauropods [32]. Food processing and buoyancy control have been discussed as the purpose of stomach stones. As Henderson [33] pointed out, these gastroliths were not required to initiate sinking in air-breathing aquatic plesiosaurs but were probably effective in minimizing instability and controlling buoyancy during submersion with compressed lungs. Thus, presumed gastroliths found in the closely related *Baryonyx* [19] might not have been ingested by chance, as suggested by Wings [34].

The position of the center of mass is important for the modes of movement and rest on land and the inclination in water [33]. In 2014, Ibrahim et al. [5] favored a position in the center of the trunk, which would result in a quadrupedal walk and a good adaptation to swimming. After the analysis of Henderson [15] moved the center back to the front of the pelvis suggesting bipedal walk and instable floating, Ibrahim et al. ([7]**,** Supplementary Data 2) presented a compromise position. However, all these considerations assumed a slightly retracted, raised head. Clearly, the straight neck of the swim posture as proposed by us [8] and in the animation related to the article by Ibrahim et al. [7] moves the center of mass forward and favors swimming, while an S-shape neck posture as reconstructed from neck vertebra [23] moves it backwards, favoring bipedal locomotion on land and corresponds to the submerged resting posture proposed by us [8] (Figure 1).

### 3.2. Hydrodynamic Aspects

The new data of Ibrahim et al. [7] permit improved experimental and numerical simulations of the hydrodynamics of *Spinosaurus*’ whole body. However, the authors focused on the swimming potential of its tail by comparing the hydrodynamic parameters of foils resembling the two-dimensional lateral shapes of the tails of *Spinosaurus*, a crested newt, a crocodile and two terrestrial theropods. When calculating thrust and efficiency, the ‘tails’ of crocodile and crested newt and even the rectangular control foil outperform the *Spinosaurus* ‘tail’, suggesting that it was less well adapted to swimming. However, if thrust and efficiency criteria measured with 20-cm foil models were applied to *Spinosaurus*, it must be asked why evolution has not ‘improved’ the *Spinosaurus* tail shape. Further, it must be asked how reasonable it is to assume that the two-dimensional, uniformly flexible, smooth foils with constant thickness possess the hydrodynamic and biomechanical properties of the three-dimensional tails of very different animals. For *Spinosaurus*, there are many unconsidered effects, for example variation in cross-sectional shape and different elastic properties and roughness along the tail or possible active tilting of tail areas around the craniocaudal axis.

If we compare for example three different objects with the same circular front faces and the shapes of a droplet, a sphere and a disk, which can be considered a foil-model of the former two shapes, their form drags are related like 1, 8 and 20. Likewise the rounded, proximal third of *Spinosaurus*’ tail as proposed by Ibrahim et al. [7] produced relatively lower hydrodynamic drag in a lateral tail strike than the much flatter base of the newt’s tail or the model foil, suggesting an even lower performance of a three-dimensional model of *Spinosaurus*’ tail. In the modeling of fish swimming, foils with non-uniform stiffness distributions are known to allow much more accurate reproductions of the kinematics than uniform foils [35]. In the undulatory propulsion of fish, fundamental questions are still open, such as the extent to which changes in the whole body stiffness affect locomotor performance and how active modulation of stiffness during propulsion and across changes in swimming speed affect propulsive speed and efficiency [36].

Ibrahim et al. [7] reconstructed a proximal-to-distal sequence of changes in the geometry of the caudal vertebra and the arrangement of major muscles. It shows three characteristic parts of the tail with different cross-sections. The proximal third of the tail is a rounded massive muscle with strong transverse processes forming a unit with the pelvic girdle without a constriction at the tail base as in the newt. While this part anatomically belongs to the tail, hydrodynamically it probably did not much contribute to propulsion. Clearly, *Spinosaurus* would have benefitted in the foil competition, if only the distal two thirds of its tail would have participated.

However, the tail must be seen in the context of the propelled body. For comparable flow characteristics of geometrically similar animals swimming under water (note that the induced surface waves make the comparison of objects swimming at the surface very complex), the ratio of viscous and inertial forces they experience must be similar. This criterion is coded in Reynolds number
(1)$$R_{e}=\frac{vl\rho }{\eta }=\frac{vl}{\upsilon }$$
where the quotient of viscosity *η* and density *ρ* gives the kinematic viscosity, which is *υ* = 0.893 × 10^−6^ m^2^/s for water at 25 °C. For the conservative assumption of *Spinosaurus* having a characteristic length of *l* = 10 m and moving at the velocity of a crocodile of *v* = 15 km/h [25], we obtain *R_e_* = 4.67 × 10^7^. For a 3-m sailfish that can travel up to 110 km/h (though with flipped-in fins, cf. [8]), we obtain similar Reynolds numbers suggesting that reasonable conclusions can be drawn when comparing the swimming behavior of these animals.

In contrast, the comparison with a 15-cm crested newt is unrealistic. Assuming the newt moving at 1.8 km/h [37]**,** *R_e_* = 8.399 × 10^4^, more than 500 times lower than a *Spinosaurus*, the relationship is comparable to that of a small rowing boat and a supertanker. This makes a comparison of the shapes of the tails propelling these animals difficult. To theoretically achieve the Reynolds number of *Spinosaurus* the newt must travel faster than 750 km/h.

For the newt, viscous surface drag that predominates at low Reynolds numbers plays an important role, stopping its propagation almost instantaneously after active tail movement ceases (cf. to video footage on swimming newts on the internet). The newt has to invest relatively little energy to overcome inertia and swims in a ‘friction mode’ under largely laminar flow conditions with a drag force increasing linearly with the swim velocity. In contrast, pure inertia effects governed the movement of *Spinosaurus* and its drag force increased approx. quadratically with the swim velocity [38]. While *Spinosaurus* had to invest more energy in overcoming inertia, it accumulated kinetic energy and could profit from longer gliding phases. Simple allometric considerations [39] show that the power that newt and *Spinosaurus* muscles have to generate increases with the square and the cubic of their cruising velocities, respectively. Accordingly, a streamlined body with a low frontal area inducing few unwanted turbulences is vital for *Spinosaurus* but not for a newt. This also imposes different conditions for the deflected, undulating tail resulting in different optimal shapes. One minor issue that should still be noted when modeling the swim is the shape of the tail tip, which appears to have been modeled without having the actual fossil remains [7].

We hypothesize that the strong cranio-caudal muscular structures along the proximal flanges of the dorsal spines represent an adaptation to swimming [7]. Such massive dorsal musculature is not normally required for quadrupedal walking or wading. However, in undulated swimming dorsal muscles must balance the abdominal muscles forces for lateral flexion of the trunk and dorsal sail and ensure the balance of ventral and dorsal forces acting on the vertebrae of the spine. This balance is necessary throughout the spine and caudal vertebrae to generate a rectilinear momentum for the entire body, as studied with robotic fish models [36]. During propulsion, it facilitates lateral flexion of the trunk and sail in concert with tail undulations, similar to flexion of the body and dorsal fin in unison with undulations of the caudal fin in carangiform fish with their strong dorsal musculature. Despite the recent discovery of the swim tail, Hone and Holtz [40] question the lifestyle of *Spinosaurus* as an aquatic predator. In their elaborate discussion of the pros and cons of *Spinosaurus’* adaptations, they use a more massive skeletal model than that published by Ibrahim et al. [5,7]. We believe that the flexibility of the skeleton of *Spinosaurus* may become an important topic in future discussions of lifestyles.

In the dorso-ventrally greatly enlarged disc-shaped proximal flanges of the dorsal spines, we see another adaptation to aquatic life. The flanges were probably more elastic to lateral deflection due to strong lateral bending forces than the rounded parts of the spines, as can be deduced from simple second moment of area considerations for these profiles. This would have allowed short-term storage of bending energy and helped to convert the lateral bending forces acting on the sail into rolling forces acting on the entire trunk. The strong cranio-caudal muscular structures along the proximal flanges of the dorsal spines provided a stable lateral base for lateral flexion of the spine and may have helped protect individual vertebrae from being twisted about the cranio-caudal axis or sheared and tilted against their neighbors (see Extended Data Figure 7 in [7]). Stability against shearing and tilting of the vertebrae was presumably also provided by the interspinal and supraspinal ligaments and overlapping zygapophyses.

To consider the achievable muscle force, power and energy consumption of the undulating tail in the hydrodynamic context of propelling the whole body, allometric relations must be applied [15,38]. They must consider the mass of body to be accelerated and mass of water to be displaced. The masses increase with *l*^3^ of geometrically similar animals, whereas the force that can be generated by the tail is proportional to the tail muscle’s cross-sectional area, which increases with *l*^2^. To partly compensate these relations, *Spinosaurus*’ tail had a relatively higher muscle volume than in the newt.

We see the muscular proximal part of the tail with its hydrodynamically negligible fin-seam as an actuator bridge between two hydrodynamically effective structures, the dorsal sail and the flat distal part of the tail. As in fish, this setup permits two swimming modes. At low speed, fishes use only their fins and produce thrust for fast swimming by undulations of the whole body [41]. For *Spinosaurus*, we propose continuous slow swimming by undulations of the distal part of the tail with its probably webbed feet helping in maneuvering, while the dorsal sail provided stability. During accelerated fast swimming, deformation of the whole body allowed the sail to contribute to thrust. During propulsion in the fast mode, the sail would have shed a vortex wake that modified the flow environment for the tail, like in fish [41]. The efficient exploitation of the hydrodynamic interaction of dorsal sail and tail through the vortex wake required delicate fine tuning of swimming speed, stroke frequency and deflection amplitudes of body and tail suggesting that the optimal swimming speed can in principle be found in model experiments. However, we assume that *Spinosaurus*’ tail shape is optimal for persistent, slow swimming in search for prey. While high thrust in accelerating bursts determines the newt’s tail shape and crocodiles need high thrust in bursting attacks to snatch on prey, *Spinosaurus* may have relied on rapid head movements.

### 3.3. Is the Head Crest an Adaptation to ‘Pivot’ Feeding?

Theropod heads are often decorated with crests, horns or rugose structures, which could have been extended by keratin and possibly aided by coloration to provide visual cues for foes, competitors or potential mates [42]. The structures may have also served for CNS cooling in large dinosaurs [13].

Up to now there has been little discussion on the head crests of spinosaurs (Figure 1). Very similar crest structures appeared millions of years before the appearance of *Spinosaurus aegyptiacus* in *Suchomimus* [43] and in the piscivorous Brazilian Spinosaurinae [44]. Although not always well preserved in fossils, the crest is found in all known spinosaurids, such as *Baryonyx* [20,43] and *Irritator* [17], which all developed aquatic adaptations although to a lesser degree than *Spinosaurus*. However, its presence in many different species over a period of more than 50 million years (cf. [7]) suggests a functional evolutionary pressure in addition to the display function proposed by Hone and Holtz [14], suggesting that head crests combined with a narrow snout were part of the piscivorous adaptations.

For the head of *Irritator*, Sues et al. [17] emphasize the differences from the long, tubular snout of *Gavialis* [45] and hypothesize the importance of the crest for the forceful biting down on prey by strengthening the narrow snout and occipital segment of the skull. By reducing the bending stress acting on the snout when impaling and holding prey, the crest would have supported the extensive secondary bony palate [46]. Sues et al. [17] proposed that spinosaurids rapidly seized smaller prey, which they processed by dorsoventral motion of the head employing their powerful neck musculature. This view is encouraged by the short cranial vertebra, which were wider than high, reducing the neck’s flexibility in lateral strikes of the head for capturing fish in a gharial-like fashion [47]. Cuff and Rayfield [48] compared the strike potentials of *Spinosaurus* and *Baryonyx* with extant crocodilians. When size-corrected, the resistance of *Spinosaurus*’ rostrum to dorsoventral bending falls between that of alligators and gharials while its resistance to mediolateral bending was lower than that of gharials. The authors concluded that *Spinosaurus* primarily captured and stunned smaller prey by dorsoventral rather than mediolateral movements.

Here, we propose an additional function of the head crest during ‘pivot’ feeding, where rapid dorsoventral head movement overcame the prey’s escape abilities following a stealthy advance. In seahorses, pivot feeding is associated with head shape adaptation minimizing hydrodynamic disturbance and reducing friction [24]. However, there are two major problems for direct transferability to *Spinosaurus*, (i) the different size of the heads of *Spinosaurus* and the seahorse, resulting in different Reynolds numbers and hydrodynamic restrictions and, (ii) the seahorse’s head is much larger than the prey, which was not the case in *Spinosaurus*. While the head morphology of the seahorse minimizes hydrodynamic disturbance above the end of the snout where feeding strikes occur, *Spinosaurus* needed to approach larger prey. This need may have selected a head shape that minimized hydrodynamic disturbance during the stealthy advance of the whole predator, with the crest possibly functioning as a cranial keel that provided stability and streamlined swimming. It may have reduced the pressure drag of the head by flow separation (defined transition to turbulence) and its wake possibly helped the dorsal sail [49]. During steady swimming, it improved the maneuverability of the head and in turn of the whole animal. If our ideas were correct, distortion measurements in a flow channel may help determining the optimal cruising speed and head posture for *Spinosaurus* when searching for prey.

## 4. Conclusions

As a rule of thumb for model considerations, additional effects can be expected if objects with size differences greater than a factor of four are compared. In whatever case, considerations of *Spinosaurus*’ swimming performance require the context of the whole body, taking into account the tail, dorsal sail and foot webbing. In our view, *Spinosaurus*’ cranial crest combined beneficial effects, such as increased snout stability, minimized hydrodynamic disturbance for slow, stealthy swimming and reduced friction for fast swimming and pivot feeding. It would be interesting to know *Spinosaurus*’ surface structures, since possibly structured scales would have modulated the surface frictional drag, as discussed for mosasaurids [49]. Surface structures typically modify the surface drag in the percentage range up 10%. They may have reduced and increased the surface drags of the body and tail to reduce friction and increase thrust, respectively. We think that maybe it is time to ask if *Spinosaurus* was semi-terrestrial, and we look forward to the full body model simulation announced in National Geographic.

## Figures and Tables

**Figure 1 life-11-00889-f001:**
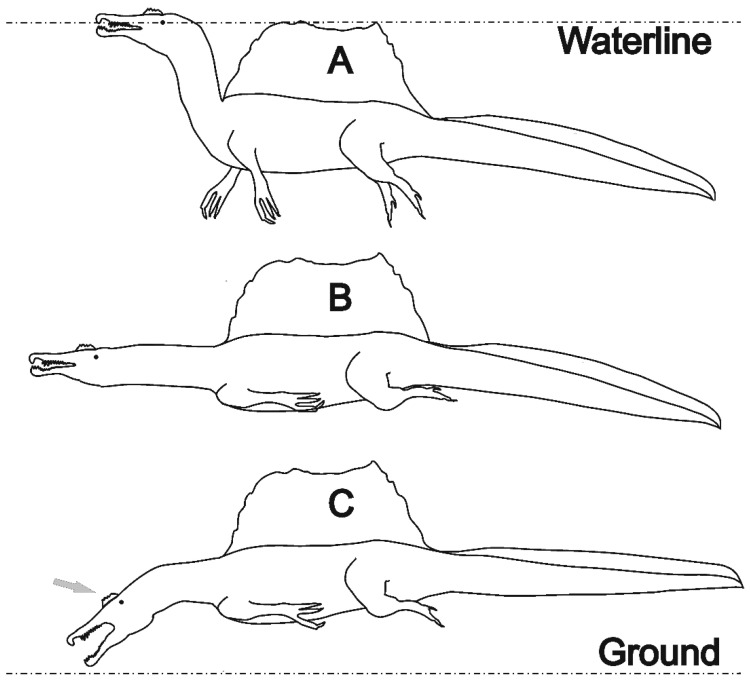
Silhouettes of *Spinosaurus* in different postures adapted from [8] and drawn in line with the skeletons in [5,7]. (**A**) Resting or stalking posture; (**B**) while cruising, frontal drag is reduced by a largely straight aligned body. The upwards oriented pitch force induced by the dorsal sail is compensated by a vertical force component generated by the slightly hanging tail. (**C**) When hunting ground-level prey like *Onchopristis*, up to about 6 m long, the pitches induced by the drag forces of dorsal sail vs. head and neck compensated. We consider the crest (arrow) as part of the adaptations associated with the specialized head and snout morphology (see text). After catching large prey, the large thumb claw may have functioned as a hook. The dorsal sail provided stability for lateral catches and in the ensuing fight.

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
