# Peer review of "Contributions to a Discussion of Spinosaurus aegyptiacus as a Capable Swimmer and Deep-Water Predator"

_life, 2021, doi:10.3390/life11090889_

Round 1

Reviewer 1 Report

In previous, the view that Spinosaurus was a capable swimmer was mostly questioned. Following the excavation of a large part of the fossil remains of the swim tail, the chances for a discussion of how Spinosaurus hunted under water are much improved.

The contribution of this study can be summarized as:

  1. The concept of crocodile-like horny scales, improving the efficiency of the undulating propulsion became obsolete for Spinosaurus, and this study offers a new explanation.
  2. This study propose that the long, flexible neck, the specialized head, and snout morphology with the ridged longitudinal fluted crest were all part of the adaptations to ‘pivot’ feeding, where rapid head movement overcomes the prey’s escape abilities as observed in seahorses.
  3. Head shape adaptation, minimizing hydrodynamic disturbance and reducing friction, would have improved hunting success, especially in turbid and low-light aquatic environments, when Spinosaurus could have used rostral integumentary mechanoreceptors for prey detection.

For this study, we have following comments:

  1. The logic of the introduction is not very clear. For example, I was puzzled by the first paragraph of the introduction. What’s the relationship between Stromer’s and Reichenbach’s studies? Why other large predators can make the areas unsafe?
  2. In Discussion, we found this study used a lot of data from previous studies. Is this study just an analysis of previous studies? If not, what’s the mathematical models used in this study. This study only used a formula for the calculation of Reynolds number, but only relying on this formula may not be convincing enough.
  3. In section 2.2 (Hydrodynamic aspects), why use a 20-cm foil models to measure the thrust and efficiency criteria?
  4. More comparative data should be included in the paper to more intuitively represent the swimming potential of their tails.
  5. Besides, I suggest that using charts to display the results may improve the readability of this study.
  6. This paper is a bit descriptive. More proofs should be added on. 

Reviewer 2 Report

The manuscript entitled “Contributions to a discussion of Spinosaurus aegyptiacus as a capable swimmer and deep-water predator” presents new findings on Spinosaurus’ swim tail basing on Reynolds number and comparison of its flow characteristics with geometrically similar animals and their swimming abilities under water.

This is an interesting article that worth’s publication after all following clarification and changes :

Queries and suggestions :

  • In line 155 this sentence “their form drags are related like 1 : 8 : 20 “need be changed to “their form drags are related like 1, 8 and  20”.

  • In lines 185-188, this sentence “Assuming the …… animals difficult” is very long and should be splitted to be more understood. The same thing can be said according to the sentence from line 251 to 254.

  • In line 199 “Simple allometric considerations show” reference should be added.

  • Figure 1 has a big legend, it is more efficient to put a small title and discuss after these positions inside the text.

  • The authors said : If our ideas were correct, distortion….. whey they don’t have done experiences on some existing animals with similar Re, or some simulations, to prove their ideas on Spinosaurus.

  • Please rearrange the paper according to the above advice. On many other occasions, the paper should be improved as well. So, please read your paper in details and rewrite it to make it more coherent.

Reviewer 3 Report

The authors do raise a few new points that suggest the possibility of swimming by Spinosaurus, but their claims are all qualitative with much arm-waving and vagueness. A quantitative analysis is the form of some basic physics, and fluid dynamics especially, is needed to support the many untested assertions presented in the text. Many more diagrams and illustrations are needed to held readers understand and assess the claims of the authors. 

Reviewer 4 Report

The authors propose new prospective on the aquatic locomotion of the dinosaur Spinosaurus. I found the manuscript interesting and novel in its approach. I have only few concerns regarding the formalization of the methods and results. See below my comments for further details. I would also suggest the authors to include a list of references that are required in my opinion. I suggest publication of the manuscript.

Comments:

In general, I would strongly suggest to create a method section in the manuscript to allow the reader to clearly understand the methods applied and the source of the data.

Line 61: it should be specified that the paper by Henderson (2018) was published before the latest reconstruction and data proposed by Ibrahim et al. (2020). This means that the study by Henderson and the criticisms put forward are not up to date. Although I believe that is important to report such criticisms, their timing should be explicitly reported in the manuscript.

Line 71: please, put a reference for each referred taxon (see below)

Line 105: can you list the “reasonable assumptions”?

Line 123: you might want to substitute “need” with “might”

Line 146: I am generally against the concept of optimization of morphological structures to ecological requirements. Taxa simply need to be good enough to fulfill mechanical requirements. This explains why there is a certain morphological variance among animals, including sister taxa, to resolve similar niche problematics. I would avoid some deterministic tone

Line 237: why “circumferences”? I see only silhouette of different poses for Spinosaurus.

Line 271: please substitute the reference for Ichtyovenator with Allain et al. Additionally, Ichtyovenator has no skull remains. Therefore, I guess you were referring to Suchomimus, Baryonyx and the Brazilian taxa, specifically Irritator.

Line 288: please, add Rayfield et al. (2007) to the list of references describing feeding in spinosaurids.

Additional references:

  • Malafaia, E., Gasulla, J. M., Escaso, F., Narváez, I., Sanz, J. L., & Ortega, F. (2020). A new spinosaurid theropod (Dinosauria: Megalosauroidea) from the upper Barremian of Vallibona, Spain: Implications for spinosaurid diversity in the Early Cretaceous of the Iberian Peninsula. Cretaceous Research106, 104221.
  • Malafaia, E., Gasulla, J. M., Escaso, F., Narvaéz, I., & Ortega, F. (2020). An update of the spinosaurid (Dinosauria: Theropoda) fossil record from the Lower Cretaceous of the Iberian Peninsula: distribution, diversity, and evolutionary history. Journal of Iberian Geology46(4), 431-444.
  • Samathi, A., Sander, P. M., & Chanthasit, P. (2021). A spinosaurid from Thailand (Sao Khua Formation, Early Cretaceous) and a reassessment of Camarillasaurus cirugedae from the Early Cretaceous of Spain. Historical Biology, 1-15.
  • Schade, M., Rauhut, O. W., & Evers, S. W. (2020). Neuroanatomy of the spinosaurid Irritator challengeri (Dinosauria: Theropoda) indicates potential adaptations for piscivory. Scientific Reports10(1), 1-9.
  • Rayfield, E. J., Milner, A. C., Xuan, V. B., & Young, P. G. (2007). Functional morphology of spinosaur ‘crocodile-mimic’dinosaurs. Journal of Vertebrate Paleontology27(4), 892-901.
  • Allain, R., Xaisanavong, T., Richir, P., & Khentavong, B. (2012). The first definitive Asian spinosaurid (Dinosauria: Theropoda) from the early cretaceous of Laos. Naturwissenschaften99(5), 369-377.

Round 2

Reviewer 1 Report

I read the revised version of the paper and make sure all the modifications meet the requirement for publication.For my part, I have no comments on it. 

Author Response

Dear reviewer,

We thank you for agreeing to the publication of our manuscript despite the critical points you might still have in mind.

Best wishes,

Jan Gimsa

Reviewer 3 Report

I do not feel that the points I raised in my first review have been dealt with in a satisfactory manner. The manuscript still needs a lot more work to make publishable. 

Author Response

Dear reviewer,

Thank you very much for your critical comment. We would like to point out again that we submitted a hypothesis manuscript and not a full paper. In the new version, we have added a new reference to the full paper "Hone, D.W.E. and Holtz, T.R., Jr. 2021. Evaluating the ecology of Spinosaurus: Shoreline generalist or aquatic pursuit specialist? Palaeontologia Electronica, 24(1): a03. https://doi.org/10.26879/1110", which contains a detailed, albeit in our opinion very critical, discussion of the pros and cons of Spinosaurus adaptations to the aquatic life as a predator. However, it may answer many of the unanswered questions that we may not have satisfactorily answered from your perspective.

Best wishes,

Jan Gimsa